# EFFICIENT SURROGATE GRADIENTS FOR TRAINING SPIKING NEURAL NETWORKS

## ABSTRACT

Spiking Neural Network (SNN) is widely regarded as one of the next-generation neural network infrastructures, yet it suffers from an inherent non-differentiable problem that makes the traditional backpropagation (BP) method infeasible. Surrogate gradients (SG), which are an approximation to the shape of the Dirac's $\delta$-function, can help alleviate this issue to some extent. To our knowledge, the majority of research, however, keep a fixed surrogate gradient for all layers, ignorant of the fact that there exists a trade-off between the approximation to the delta function and the effective domain of gradients under the given dataset, hence limiting the efficiency of surrogate gradients and impairing the overall model performance. To guide the shape optimization in applying surrogate gradients for training SNN, we propose an indicator $\chi$, which represents the proportion of parameters with non-zero gradients in backpropagation. Further we present a novel $\chi$-based training pipeline that adaptively makes trade-offs between the surrogate gradients' shapes and its effective domain, followed by a series of ablation experiments for verification. Our algorithm achieves 69.09% accuracy on the ImageNet dataset using SEW-ResNet34 - a 2.05% absolute improvement from baseline. Moreover, our method only requires extremely low external cost and can be simply integrated into the existing training procedure.

## 1 INTRODUCTION

Spike Neural Networks (SNN) have gained increasing attention in recent years due to their biological rationale and potential energy efficiency as compared to the common real-value based Artificial Neural Networks (ANN). SNN communicates across layers by the addition of spiking signals. On the one hand, this spiking mechanism turns multiplicative operations to additive operations, increasing the inference procedure's efficiency. On the other hand, it introduces an intrinsic issue of differentiability, which makes training SNNs more challenging. At present, the method for obtaining practical SNNs can be roughly divided into three categories: converting a pretrained ANN to SNN (Sengupta et al., 2019; Deng & Gu, 2020; Li et al., 2021a; Bu et al., 2021), training with biological heuristics methods (Hao et al., 2020; Shrestha et al., 2017; Lee et al., 2018), and training with BP-like methods (Wu et al., 2018; Zheng et al., 2020; Li et al., 2021b; Yang et al., 2021). The converting method may not promote increased inference efficiency in practice since it requires a lengthy simulation period (high inference latency) to catch up to the accuracy of the source ANN (Sengupta et al., 2019; Rueckauer et al., 2017). Although the biological heuristics technique requires just local information to change network parameters, it is confined to small datasets due to its limitation in representing global information (Wu et al., 2018; Shrestha et al., 2017). Compared to these two approaches, direct training with BP-like method is capable of handling complex models with a very short simulation duration to attain adequate model performance (Zheng et al., 2020; Fang et al., 2021; Li et al., 2021b).

With the help of surrogate gradient, the SNN can be directly trained through the BPTT algorithm on an ANN-based platform (Wu et al., 2018). Nonetheless, there is a non-negligible performance disparity between directly trained SNN and ANN, particularly on large and complicated datasets (Deng et al., 2020; Jin et al., 2018). This is because training SNN with surrogate gradient can only obtain approximate gradients, and the final performance is highly affected by the surrogate gradient shape. A more suitable surrogate gradient shape usually results in a better performing SNN (Neftci et al., 2019). However, an appropriate surrogate gradient must strike a compromise between the approximation shape and the effective domain of gradients. So just altering the shape of the

surrogate gradient to be more similar to the $\delta$-function may result in the training failing due to gradient disappearance, as the gradients of most membrane potentials are extremely small. Additionally, the optimal surrogate gradient shapes for various layers may different and may change throughout the training process (Li et al., 2021b). As a result, using a fixed initial surrogate gradient shape (adequate effective domain) during the whole training phase will always have a substantial gradient error, which affects the final training result.

The purpose of this work is to optimize the SNN training pipeline by adaptively altering the shape of surrogate gradient in order to control the effective domain for the surrogate gradients. We suggest an index $\chi$ to denote the proportion of membrane potential with non-zero gradients in backpropagation and present a technique to control the proportion of non-zero gradients (CPNG) in the network. The CPNG technique modifies the shape of surrogate gradients during network training, progressively approaching the $\delta$-function while maintaining the index $\chi$ steady within an effective range to ensure training stability. Finally, each layer succeeds in finding a surrogate gradient shape that makes a better balance between the approximation error to the $\delta$-function with the size of effective domain than the fixed-shape surrogate gradients. It's worth mentioning that our strategy only incurs minor additional costs during the training phase and has no effect on the inference phase. We verify the compatibility of CPNG to the existing mainstream SNN infrastructures such as VGG (Simonyan & Zisserman, 2014), ResNet (He et al., 2016), and Sew-ResNet (Fang et al., 2021). In all reported comparative experiments, training with CPNG gives more accurate models than training with vanilla surrogate gradients.

Our main contributions can be summarized as follows:

- We identify and investigate the impact of the shape of surrogate gradients on SNN training. Our finding characterizes a special representative power for SNN that can be utilized to improve its performance.

- We propose a statistical indicator $\chi$ for the domain efficiency of surrogate gradients and a $\chi$-based training method CPNG that adjusts the shape of surrogate gradients through the training process, driving the surrogate gradients close to the theoretical $\delta$-function with ensured trainability on sufficiently large domains.

- Our CPNG method improves classification accuracy on both static image datasets including CIFAR10, CIFAR100 and ImageNet, as well as event-based image datasets such as CIFAR10-DVS. We achieve an accuracy of 69.09% in the experiment that trains ImageNet on Sew-ResNet34.

## 2 RELATED WORK

There are two primary branches of training a high-performing deep spiking neural network, converting a pretrained artificial neural network to its corresponding spiking neural network, and directly training a spiking neural network through BP-like method.

**ANN-SNN Conversion**  ANN-SNN conversion takes advantage of the high performance of ANN and converts the source ANN to the target SNN through weight-normalization (Diehl et al., 2015; 2016) or threshold balancing (Sengupta et al., 2019). However, SNN forming this method requires a huge simulation length to catch up with the source ANN's performance. Numerous strategies have been proposed to shorten the simulation time, including robust threshold (Rueckauer et al., 2016), SPIKE-NORM (Sengupta et al., 2019), and RMP (Han et al., 2020). A work (Deng & Gu, 2020) examines the conversion error theoretically, decomposes it layer by layer, and offers threshold ReLU and shift bias procedures to decrease the error. Based on it, Li et al. (Li et al., 2021a) divide the conversion error into clip error and floor error and design adaptive threshold, bias correction, potential correction, and weight calibration to dramatically decrease the required simulation length. A recent work (Bu et al., 2021) further proposes unevenness error, trains ANN with a novel activation function and reduce simulation length.

**BP-like Method**  HM2-BP (Jin et al., 2018) enables SNN to adjust the spike sequence rather than just the spike at a certain moment. TSSL-BP (Zhang & Li, 2020) decomposes the backpropagation error into inter and intra interactions, calculating the derivatives only at the spiking moment. NA algorithm (Yang et al., 2021), which calculates the gradient of the non-differentiable part through

finite difference. The surrogate gradient based BP algorithm uses differentiable functions instead of the Dirac's $\delta$-function to mitigate the non-differentiation problem in the process of SNN training. Different from ANNs, SNNs naturally have time attribute. Therefore, the existing studies consider both temporal and spatial information in BP procedure (Wu et al., 2018). Normalization methods are crucial in SNN training, which help speed up the network's convergence and prevent gradient disappearance or explosion. For this purpose, SNN-friendly normalization algorithms such as NeuNorm (Wu et al., 2019) and threshold-dependent batch normalization (tdBN) (Zheng et al., 2020) have been developed. The majority of studies have employed fixed-shape surrogate gradients, and some work has preliminarily explored the performance of surrogate gradients with different shapes (Wu et al., 2018; Neftci et al., 2019; Bellec et al., 2018). A recent work looked into the shape-changing surrogate gradient (Li et al., 2021b), proposed Dspike as surrogate gradient, and used the finite difference to guide the change of Dspike's shape, significantly improving the performance of SNN. In order to train a very deeper SNN, Sew-ResNet (Fang et al., 2021) structure is proposed, which enables SNN training even on a 152-layer network.

## 3 PRELIMINARY

Through out the paper, we use bold letters to denote matrices and vectors, superscripts to identify specific layers, subscripts to denote specific neurons, and indexes to identify specific moments.

**Leaky Integrate-and-Fire Model.** We use the Leaky Integrate-and-Fire (LIF) module for spiking neurons. Formally, given the pre-synaptic input (denoted by $c_i^{(l)}[t+1]$) of the $i^{th}$ neuron in the $l^{th}$ layer at time step $t + 1$, we can model the iterative process in LIF as

$$c_i^{(l)}[t+1] = \sum_{j}^{N^{(l-1)}} w_{ij}^{(l)} s_j^{(l-1)}[t+1], \tag{1}$$

$$u_i^{(l)}[t+1] = \tau u_i^{(l)}[t](1 - s_i^{(l)}[t]) + c_i^{(l)}[t+1], \quad s_i^{(l)}[t+1] = H(u_i^{(l)}[t+1] - \nu). \tag{2}$$

Here, $N^{(l-1)}$ is the number of neurons in the $(l-1)^{th}$ layer, $s_j^{(l-1)}[t+1]$ is the output spike of the $j^{th}$ neuron in the $(l-1)^{th}$ layer at time $t+1$, $w_{ij}^{(l)}$ is the weight between $j^{th}$ neuron in $(l-1)^{th}$ layer and $i^{th}$ neuron in $l^{th}$ layer, $u_i^{(l)}[t]$ is the membrane potential of the $i^{th}$ neuron in the $l^{th}$ layer at time $t$, $\tau$ is the membrane potential attenuation factor, $H(\cdot)$ is the step function, and $\nu$ is the activation threshold. When the membrane potential of a neuron exceeds the activation threshold, a spike is released and the membrane potential of the current neuron is set to zero.

**Surrogate Gradient Function.** There are various surrogate gradient shapes adopted by previous work (Wu et al., 2018; Neftci et al., 2019). In this work, we used triangle-like function, rectangular-like function and arctan-like function to verify the effectiveness of CPNG. These functions are described below:

$$f_{\text{triangle}}(x) = \begin{cases} \beta(1 - \beta|x - \nu|) & \text{if} \quad |x - \nu| < 1/\beta \\ 0 & \text{otherwise} \end{cases}, \tag{3}$$

$$f_{\text{rectangular}}(x) = \begin{cases} \beta & \text{if} \quad |x - \nu| < 1/(2\beta) \\ 0 & \text{otherwise} \end{cases}, \quad f_{\text{arctan}}(x) = \frac{\beta}{1 + (\pi\beta(x - \nu))^2}, \tag{4}$$

where $\beta$ represents the maximum gradient value of current surrogate gradient function. Notably, the surrogate gradient satisfy $\int_{-\infty}^{+\infty} f(x) = 1$, which is also the property of the $\delta$-function.

**Loss Function.** In our experiments, we use cross-entropy-loss ($\mathcal{L}_{CE}$) as the loss function. The formula is given as

$$\text{out} = \frac{1}{T} \sum_{t=1}^{T} \mathbf{W}^{(L)} \cdot \mathbf{S}^{(L-1)}[t], \quad \text{Loss} = \mathcal{L}_{CE}(\text{out}, \text{label}), \tag{5}$$

where $T$ represents the simulation time, $L$ represents the last layer of the SNN, $\mathbf{W}^{(L)}$ represents the weight matrix of the $L^{th}$ layer, and $\mathbf{S}^{(L-1)}$ represents the output spike vector of the $(L-1)^{th}$ layer. Consistent with (Wu et al., 2019; He et al., 2016), our final output layer is a voting layer devoid of any LIF model.

## 4 METHOD

### 4.1 SHAPE PARAMETERS AND EFFECTIVE DOMAIN INDICATOR

**Shape Parameters.** Intuitively, increasing the shape parameter $\beta$ of surrogate gradients would drive it closer to the $\delta$-function (Fig. 1(right)). One might expect to adopt a very high $\beta$ to obtain SNN for a good performance. We first examine whether this intuitive approach is possible. We trained VGG16-structured SNN (replace ReLU with LIF, and use average pooling.) on CIFAR100 using triangle-like surrogate gradient with $\beta$ set from 0.25 to 2 respectively. As shown in Fig. 1(left), the test accuracy increases when $\beta$ varies from 0.25 to 1.0 but remains at 1.0% when $\beta$ is set to 1.5 and 2.0, indicating that properly increasing $\beta$ may benefit the training but arbitrarily increasing $\beta$ will drive the training collapse. According to another viewpoint (Zenke & Vogels, 2021; Herranz-Celotti & Rouat, 2022), when the beta is greater than 1, the gradient explosion on the deep SNN will occur, resulting in training failure; thus, some of the most recent surrogate gradients fix the maximum value of surrogate gradients at 1 (Suetake et al., 2022; Zenke & Ganguli, 2018). But the narrow effective interval of the surrogate gradients is the primary culprit in our experiments. We detected the maximum gradient absolute values for SNN training at the first 100 batches when $\beta = 1(1.0001)$ and $\beta = 1.5(1.000)$, respectively. These results illustrate that when $\beta$ is 1.5, the gradient explosion does not always occur in the network.

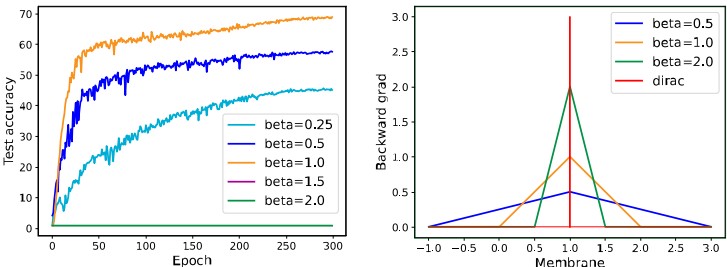

Figure 1: Left: Test accuracy of different $\beta$ when threshold is 1.0. Right: $\delta$-function and triangle-like surrogate gradient when the threshold is 1.0.

In fact, these results unveil that efficient training of SNNs requires not only the approximation to the $\delta$-function but also the insurance for the surrogate gradients to work. Thus it is necessary to employ a dynamic shape-changing strategy rather than using a fixed-shape surrogate gradient. This issue is also covered by (Li et al., 2021b) as well and was owed to the lack of adaption to the dataset variation.

**Effective Domain Indicator.** As aforementioned, if we unrestrictedly increase the $\beta$ in order to get the surrogate gradient closer to the $\delta$-function, the training curve will remain flat without any update. This is caused by the fact that the majority of the membrane potential remains outside the effective region. On the other hand, if $\beta$ is too small, it will also lead to a suboptimal training outcome due to the presence of a substantial gradient error (Fig. 1(left)) between the adopted surrogate gradients and the $\delta$-function. Thus a *proper* surrogate gradients should maintain an optimal balance between the domain effectiveness and the $\delta$-function approximation.

To quantitatively guide the choice of $\beta$, we need a statistical indicator to denote the percentage of membrane potentials that fall into the domain of surrogate gradients. As illustrated in Fig. 2(left), the distribution of membrane potentials on each layer takes the normal shape. Thus, for the simplicity of calculation, we regard the membrane potential distribution of all neurons within the same layer as a Gaussian one. By calculating the mean $\mu$ and the standard deviation $\sigma$ of the membrane potential before this layer releases spikes, we can obtain the proportion of the neuron with a non-zero gradient

during a certain iteration (the area between the red lines in Fig. 2(right)). For a given $\beta$, the effective gradient domain of triangle-like surrogate gradient is $[\nu - 1/\beta, \nu + 1/\beta]$, we can obtain the definite integral of the current normal distribution in this effective gradient domain, which is the *Effective Domain Indicator* $\chi$:

$$\chi = \int_{\nu-1/\beta}^{\nu+1/\beta} \frac{1}{\sqrt{2\pi}\sigma} e^{-\frac{(x-\mu)^2}{2\sigma^2}} dx. \tag{6}$$

For each layer, we record the membrane potential of all neurons in every time step (a tensor shaped like batchsize-by-timestep-by-channels-by-H-by-W) and calculate the mean and variance. Based on this indicator $\chi$, we can then effectively determine to what extent we can tune the $\beta$ while ensuring that there are enough membrane potentials located within the effective range of surrogate gradients to make the training progress.

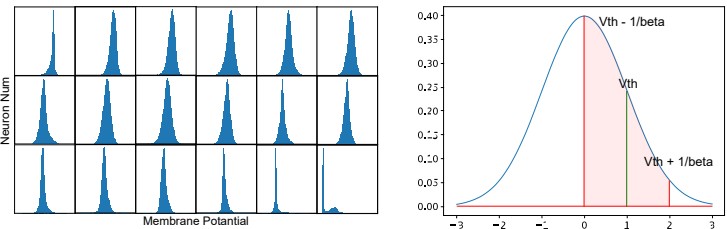

Figure 2: Left: Membrane distribution of each layer in experiment training ResNet19 on CIFAR10. Right: When the threshold is 1.0 and $\beta$ is 1.0, the proportion of neurons with non-zero gradient.

## 4.2 CPNG METHOD

In this section, we will cover how to combiningly optimize $\beta$ and $\chi$ to maximize the effectiveness of surrogate gradients. To train the network successfully, there must be sufficient membrane potential values in the effective domain of the surrogate gradient, i.e., $\chi$ must be large enough. The most extreme case is $\beta \to 0$, which gives $\chi \to 1$. Obviously, this is not an optimal solution as it introduces substantial error for the gradients. In order to determine the choice of $\beta$ for a given $\chi$, we need to investigate the gradient form $\frac{\partial \mathcal{L}_{CE}}{\partial \mathbf{S}^{(l)}[t]}$ to see how the $\beta$ affects the error back-propagation.

When $l = L - 1$, we can directively get $\frac{\partial \mathcal{L}_{CE}^{(1)}}{\partial \mathbf{S}^{(L-1)}[t]}, t = 1, 2, \cdots, T$ according to Eqn. 5.

When $l = 0, 1, \cdots, L - 2$, we can first derive $\frac{\partial u_i^{(l)}[t+1]}{\partial s_i^{(l)}[t]} = -\tau u_i^{(l)}[t]$ and $\frac{\partial u_i^{(l)}[t+1]}{\partial u_i^{(l)}[t]} = \tau(1 - s_i^{(l)}[t])$ from Eqn. 2, and further have

$$\frac{\partial \mathcal{L}_{CE}}{\partial \mathbf{u}^{(l)}[t]} = \begin{cases} \frac{\partial \mathcal{L}_{CE}}{\partial \mathbf{s}^{(l)}[t]} \frac{\partial \mathbf{s}^{(l)}[t]}{\partial \mathbf{u}^{(l)}[t]} & t = T \\ \frac{\partial \mathcal{L}_{CE}}{\partial \mathbf{s}^{(l)}[t]} \frac{\partial \mathbf{s}^{(l)}[t]}{\partial \mathbf{u}^{(l)}[t]} + \frac{\partial \mathcal{L}_{CE}}{\partial \mathbf{u}^{(l)}[t+1]} \cdot \tau(1 - \mathbf{s}^l[t]) & t = 1, 2, \cdots, T - 1 \end{cases}, \tag{7}$$

$$\frac{\partial \mathcal{L}_{CE}}{\partial \mathbf{s}^{(l)}[t]} = \begin{cases} \frac{\partial \mathcal{L}_{CE}^{(1)}}{\partial \mathbf{s}^{(l)}[t]} & t = T \\ \frac{\partial \mathcal{L}_{CE}^{(1)}}{\partial \mathbf{s}^{(l)}[t]} + \frac{\partial \mathcal{L}_{CE}}{\partial \mathbf{u}^{(l)}[t+1]} \cdot (-\tau \mathbf{u}^l[t]) & t = 1, 2, \cdots, T - 1 \end{cases}. \tag{8}$$

by Eqn. 2. Here, $\mathbf{u}^{(l)}[t]$ represents the membrane potentials vector of the $l^{th}$ layer, $\frac{\partial \mathcal{L}_{CE}^{(1)}}{\partial \mathbf{s}^{(l)}[t]}$ is the gradient directly obtained from previous layer. The complete $\frac{\partial \mathcal{L}_{CE}}{\partial \mathbf{s}^{(l)}[t]}$ needs to consider the dependence between the current moment spike and the membrane potential at the next moment (Eqn. 8). The term $\frac{\partial \mathcal{L}_{CE}}{\partial \mathbf{u}^{(l)}[t+1]}$ can be obtained iteratively from the gradients from its succeeding layer, and $\frac{\partial \mathbf{s}^{(l)}[t]}{\partial \mathbf{u}^{(l)}[t]}$ is given by surrogate gradients. We can conclude that each item of the $\frac{\partial \mathcal{L}_{CE}}{\partial \mathbf{u}^{(l)}[t]}$ contains $\frac{\partial \mathbf{s}^{(l)}[t]}{\partial \mathbf{u}^{(l)}[t]}$, while effective interval of surrogate gradient ($\chi$) determines $\frac{\partial \mathbf{s}^{(l)}[t]}{\partial \mathbf{u}^{(l)}[t]}$, which in turn affects the calculation

---

**Algorithm 1** Control the Proportion of Non-zero Gradient

---

    **Input:** SNN model with $L$ layer, current iterator epoch e,
    $\chi_{\text{limit}}$, and vector $\chi_{\text{recorder}}$: store each layer's smallest $\chi$
    **Output:** Each layer's surrogate gradient parameter $\beta$
    **if** e == 0 **then**
        **for** $l = 1, 2, \cdots L$ **do**
            calculate current $\chi$ by Eqn.6 for layer-$l$ and store at $\chi_{\text{recorder}}[l]$
        **end for**
    **else**
        **for** $l = 1, 2, \cdots L$ **do**
            calculate current $\chi_{\text{cur}}$ by Eqn.6 for layer-$l$
            **if** $\chi_{\text{recorder}}[l] < \chi_{\text{limit}}$ **then**
                $\chi_{\text{recorder}}[l] = \chi_{\text{limit}}$
            **else if** $\chi_{\text{cur}} < \chi_{\text{recorder}}[l]$ **then**
                $\chi_{\text{recorder}}[l] = \chi_{\text{cur}}$
            **end if**
            $\chi_{\text{min}} = \chi_{\text{recorder}}[l]$
            **if** $\chi_{\text{min}} \neq \chi_{\text{cur}}$ **then**
                use $\chi_{\text{min}}$ to update $\beta$ using binary search method.
            **end if**
        **end for**
    **end if**
    **return** $\beta$ for each layer

---

of $\frac{\partial \mathcal{L}_{CE}}{\partial \mathbf{W}^{(l)}}$ and $\frac{\partial \mathcal{L}_{CE}^{(1)}}{\partial \mathbf{s}^{(l-1)}[t]}$. However, the surrogate gradient is only an approximation of $\delta$-function, and an overly relaxed surrogate is harmful to the network's final performance. If we reasonably restrict the effective interval of the surrogate gradients, it is possible to drive the SNN to select those more essential membrane potentials for backpropagation.

We also need to ensure that the new $\chi$ does not make the network difficult to train, for this, CPNG sets the target $\chi$ of each layer to the smallest $\chi$ that has occurred in the current layer during the training iteration, rather than an artificial goal. If the network can be trained when the smallest $\chi$ appears, then the network should still be trained after we adjust the $\beta$ and maintain the smallest $\chi$. When using CPNG, we expect that the network parameters are appropriate, that is, the network has traversed the whole dataset to prevent the misleading of network parameters by data randomness. For example, if we use CPNG once per batch, the network parameters are mostly affected by the first few batches in the early stages of network training, and the statistical indicator $\chi$ obtained by using such network parameters will have a lot of randomnesses.

CPNG computes the smallest $\chi$ of each layer during the iteration process as $\chi_{\text{min}}$ and records it. If the $\chi$ value of a certain layer rises after an epoch, CPNG adjusts the $\chi$ value of the current layer to $\chi_{\text{min}}$ by increasing the $\beta$, otherwise, keep the current $\beta$ fixed and update $\chi_{\text{min}}$. Since the $\chi_{\text{min}}$ of each layer of neurons may be different, different layers may have different surrogate gradient shapes. In addition, we set a safe lower bound $\chi_{\text{limit}}$. When $\chi$ falls below the lower bound, the $\beta$ may decrease to force $\chi$ back above $\chi_{\text{limit}}$ in order to guarantee that sufficient membrane potential values are covered in the effective domain of surrogate gradient for successful SNN training. The CPNG algorithm is detailed in Algo.1.

### 4.3 The Cost of CPNG Method

The extra cost of CPNG occurs in two steps: (1) collecting the mean and variance of the membrane potential before releasing the spikes of each layer; (2) altering the $\beta$ using the indicator $\chi$. In our experiment, we only use the mean and variance of the last batch to calculate the indicator $\chi$, which makes the cost of the first step in the same order of magnitude as the batch normalization (Ioffe & Szegedy, 2015) operation. For the latter step, we provide a binary search method that solves the problem very fast, and further optimization algorithms can further improve the solution speed. The above analysis is the cost of using CPNG once, the overall cost takes into account the frequency of using CPNG. In our experiments, we employ CPNG just once every epoch, which is quite economical

Table 1: The impact of using time and batch size on Vgg16+CIFAR100 experiment.

|  | 256(w/o CPNG) | 256(B) | 256(E) | 512(w/o CPNG) | 512(B) | 512(E) |
|------|-----------|---------|---------|-----------|---------|---------|
| Acc | 69.08% | 71.05% | 71.54% | 68.44% | 69.59% | 70.20% |
| Time | 705.59m | 799.41m | 716.67m | 377.76m | 424.49m | 381.79m |

B: Use CPNG per batch.                     E: Use CPNG per epoch (last batch)

when compared to the network training time. Quantitatively, in the VGG16+CIFAR100 experiment, it takes an average of 1.7GFLOPs to obtain the output corresponding to an input without CPNG, and the first step of CPNG will only add $5.59 \times 10^{-3}$ additional GFLOPs. Using CPNG once per epoch takes an average of 3.3 seconds of overhead (1.57% of total training time).

## 5 EXPERIMENT

To verify the effectiveness of the CPNG method, we provide groups of comparative experiments (Sec. 5.2) on both static and neuromorphic datasets. We also compare CPNG with existing works in Sec. 5.3 and show the final $\beta$ of SNN after training with CPNG in Sec. 5.4.

### 5.1 IMPLEMENTATION DETAILS

All the SNN architectures include the tdBN layer (Zheng et al., 2020) with the average-pooling layer, and compared to their ANN versions, we replace the activation function ReLU with LIF. Our experiment settings, such as optimizer, learning rate, are detailed in Appendix. A.2. Except for applying the CPNG method at the end of each epoch, all other conditions, such as learning rate, batch size, etc., are consistent. The data preprocessing for each dataset included in the experiments are as follows:

**CIFAR and ImageNet**. We use standard processing methods for these datasets, see appendix. A.1 for more details.

**CIFAR10-DVS**. CIFAR10-DVS is an event-based image datasets. Following the previous work (Li et al., 2021b), we re-sample 10 simulation length, divide the dataset into 9k training images and 1k test images, reduce the spatial resolution to 48×48, and apply data augmentation techniques.

Table 2: $\chi_{\text{limit}}$ Experiments on ResNet18+CIFAR-DVS.

| $\chi_{\text{limit}}$ | 0.05 | 0.1 | 0.2 | 0.3 | 0.4 | 0.5 | 0.6 |
|------|------|------|------|------|------|------|------|
| Accuracy | 76.6% | 76.6% | 76.37% | 76.8% | 76.0% | 76.1% | 74.1% |

### 5.2 PERFORMANCE IMPROVEMENT WITH CPNG

Although $\chi$ is employed as a statistical indicator, table1 demonstrates that increasing the batch size does not always result in better performance of CPNG. This is because there is no strict positive correlation between neural network performance and batch size. We don't need to increase batch size excessively to increase the sample size of $\chi$. Besides, as we mentioned in 4.2, using CPNG once per batch does not guarantee performance improvement, it may bring more overhead and worse performance. Then we compared the impact of different $\chi_{\text{limit}}$ on CPNG, table2 shows that we may don't need to focus too much on $\chi_{\text{limit}}$, $0.05 - 0.2$ might be suitable. In our experiments, we set $\chi_{\text{limit}}$=0.2 uniformly.

Additionally, we tested the applicability of CPNG to various surrogate gradient functions. For surrogate gradients with non-zero gradient everywhere, such as arctan, we directly use $[\nu - 1/\beta, \nu + 1/\beta]$ as the integration interval to calculate the indicator $\chi$, then use Algo.1 to solve new $\beta$. Considering the case of using triangle-like surrogate gradient, we control the percentage of membrane potentials with non-zero gradients, which can also be explained as: under the current surrogate gradient, find a representative interval related to $\beta$, and make the calculated $\chi$ in this interval equal to the $\chi_{\text{target}}$. In the

Table 3: Examine CPNG on various surrogate gradients.

| Dataset | Method | Architecture | Time Step | Accuracy |
|---|---|---|---|---|
| CIFAR10-DVS | Triangular | ResNet18 | 10 | 75.6% |
| | Triangular+CPNG | ResNet18 | 10 | 76.37$\pm$1.02 % |
| | Rectangular | ResNet18 | 10 | 74.7% |
| | Rectangular+CPNG | ResNet18 | 10 | 75.3% |
| | ArcTan | ResNet18 | 10 | 67.2% |
| | ArcTan+CPNG | ResNet18 | 10 | 67.5% |
| | Triangular+TET | ResNet18 | 10 | 79.9% |
| | Triangular+TET+CPNG | ResNet18 | 10 | **82.3%** |

Table 4: Result of training spiking neural network.

| Dataset | Method | Architecture | Time Step | Accuracy |
|---|---|---|---|---|
| CIFAR10 | STBP-tdBN (Zheng et al., 2020) | ResNet19 | 6 | 93.16% |
| | ANN-to-SNN (Li et al., 2021a) | ResNet20 | 32 64 128 | 94.78% 95.30% 95.42% |
| | ANN-to-SNN (Bu et al., 2021) | ResNet20 | 8 16 32 | 89.55% 91.62% 92.24% |
| | CPNG | ResNet19 | 6 | **94.02**$\pm$0.05% |
| CIFAR100 | Diet-SNN (Rathi & Roy, 2020) | VGG16 | 5 | 69.67% |
| | ANN-to-SNN (Li et al., 2021a) | VGG16 | 32 64 128 | 73.55% 76.64% 77.40% |
| | ANN-to-SNN (Bu et al., 2021) | VGG16 | 8 16 32 | 73.96% 76.24% 77.01% |
| | CPNG | VGG16 | 5 | **71.32**$\pm$0.20% |
| CIFAR10-DVS | Dspike (Li et al., 2021b) | ResNet18 | 10 | 75.40% |
| | TET(Deng et al., 2022) | VGG11 | 10 | 83.17% |
| | CPNG | ResNet18 | 10 | 76.37%$\pm$0.26 |
| | CPNG + TET | VGG11 | 10 | **83.83**$\pm$0.02% |
| ImageNet | ANN-to-SNN (Li et al., 2021a) | ResNet34 | 32 64 128 | 64.54% 71.12% 73.45% |
| | ANN-to-SNN (Bu et al., 2021) | ResNet34 | 16 32 64 | 59.35% 69.37% 72.35% |
| | Sew-ResNet (Fang et al., 2021) | Sew-ResNet34 | 4 | 67.04% |
| | TET (Deng et al., 2022) | Spiking-Sew-ResNet34 | 6 | 64.79% |
| | TET (Deng et al., 2022) | Sew-ResNet34 | 4 | 68.00% |
| | CPNG | Sew-ResNet34 | 4 | **69.09%** |

case of using surrogate gradients with infinite non-zero gradient interval, the representative interval for calculating the indicator is $[\nu - 1/\beta, \nu + 1/\beta]$, if we need a smaller interval $[\nu - r_1, \nu + r_1]$ with the current mean and standard deviation, just set the new $\beta$ to $1/r_1$. The experimental results in table3 also demonstrate the effectiveness of our approach (More results are shown in Appendix A.6).

Finally, we verify the compatibility of CPNG with existing direct training methods TET (Deng et al., 2022)(table3), we got a 2.4% improvement on the ResNet18+CIFAR-DVS experiment.

## 5.3 COMPARISON TO EXISTING WORKS

In this section, the experimental results we report all use triangle-like surrogate gradient. On some datasets, the current SOTA conversion method performs better than direct training, but they need lengthy simulation time steps, especially on the ImageNet dataset. In addition, the conversion-based method cannot be applied to neuromorphic datasets such as CIFAR10-DVS. Hence, it's also necessary to design a surrogate gradient search method to optimize the direct training.

**VGG16 + CIFAR100.** We train the SNN for 300 epochs on 1 TITAN, with the batch size of 256 and the time step of 5. As shown in Table .6, CPNG manages to improve 2.24% accuracy. To show that CPNG effectively controls the rise of $\chi$, we report the $\chi$ during training in Fig. 3. The result of ResNet20+CIFAR100 is also shown in Table .6.

**ResNet19 + CIFAR10.** We use ResNet19 (Zheng et al., 2020) in accordance with tdBN layers, the batch size of 256 and the time step of 6. Then we train the SNN for 200 epochs on 4 TITAN, CPNG achieves an increase in test accuracy to 94.02%, while its counterpart is 93.26% (Table .6).

**ResNet18 + CIFAR10-DVS.** We train the model for 200 epochs on 2 TITAN Xp cards, with the time step of 10 and the batch size of 72. The CPNG obtains a higher test accuracy of 76.37% while

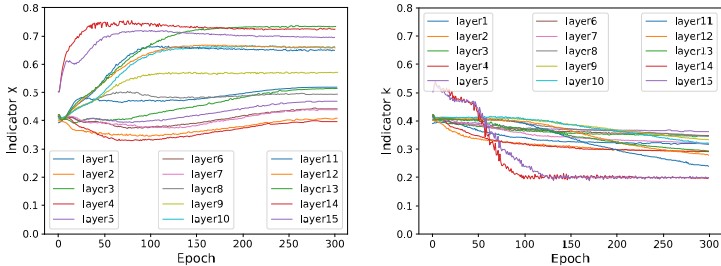

Figure 3: In the experiment of VGG16, the proportion of non-zero gradient membrane potentials of neurons in different layers without CPNG (left) and with CPNG (right).

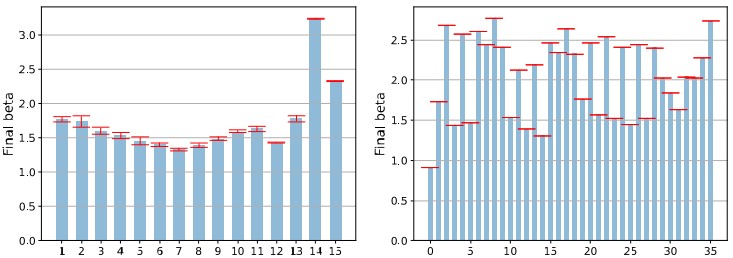

Figure 4: Left: Final $\beta$ of each layer in VGG16 experiment, the initial $\beta$ is 1.0. Right: Final $\beta$ of each layer in Sew-ResNet34 experiment, the initial $\beta$ is 1.0..

consuming significantly less overhead than Dspike (Li et al., 2021b). In addition, the combination of CPNG and TET can achieve an average test accuracy of 83.83%, which is a new SOTA.

**Sew-ResNet34 + ImageNet.** We use the same structure and membrane decay rate, etc. as Sew-ResNet (Fang et al., 2021). We train the model for 160 epochs on 8 GTX 3090 cards with a time step of 4 and the batch size of 544. Using only CPNG can achieve an accuracy of 69.04%, surpassing the current direct training SOTA.

## 5.4 Surrogate Gradient Shapes of Different Layers

We show the final $\beta$ of some experiments in Fig.4, and all experimental results are presented in the appendix A.3. Various layers' $\beta$ are different, which demonstrates that various layers match distinct surrogate gradient shapes as a result of their varying membrane potential distributions.

CPNG eventually increases the $\beta$ of most layers. Compare to CPNG, randomly increasing the $\beta$ can make the network difficult to train (Fig. 1(left)). Even with $\beta$ set to 1.5, which most of the neuron layers shown in Fig. 4(left) can approach or reach, the network is still difficult to train. This demonstrates that it is safe to increase $\beta$ using CPNG, while it is unsafe to increase $\beta$ arbitrarily.

## 6 Conclusion

This work proposes a new perspective for directing the shape change of the surrogate gradient, we propose a statistical indicator $\chi$ that guides the shape change of the surrogate gradient, and propose the CPNG method for modifying the shape of the surrogate gradient during training while guaranteeing the proportion of membrane potential with non-zero gradients. It's possible that the failure to produce satisfactory results when pulling surrogate gradient directly to $\delta$-function is due to a failure to meet the premise that the network can be trained normally. In other words, there may exists a trade-off between the approximation to the $\delta$-function and the effective domain of gradients under the given dataset, and CPNG helps us approach the equilibrium point.

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

## A  APPENDIX

### A.1  DATA PROCESSING

**CIFAR**. For the training set, we randomly crop the image to (32,32) and apply a random horizontal flip. Then all the images will be normalized to a standard normal distribution.

**ImageNet**. For the training set, we randomly crop the image to (224,224) and apply a random horizontal flip. Then for the test set, we resize the image to (256,256) and center crop the image to (224,224). Finally, all the images are normalized to a standard normal distribution.

### A.2  MORE EXPERIMENT DETAILS

In this section, we provide more experimental details. We use 1/20 of the training epochs for warm-up, linearly increase learning rate from $0.1lr$ to $lr$, then use cosine decay to reduce learning rate to 0 in the remaining epochs. In the last batch, the mean and variance of the membrane potential are additionally stored in the forward process, and after the backward process, the stored mean and variance are used to obtain a new $\beta$ according to Algorithm 1. Hyperparameters are shown in the table 5.

Table 5: Experiment Setting

| Experiment | CIFAR100 | CIFAR10 | CIFAR-DVS | ImageNet | CPNG + TET |
|---|---|---|---|---|---|
| learning rate | 0.1 | 0.1 | 0.01 | 0.01 | 0.001 |
| weight decay | 1e-4 | 1e-4 | 4e-5 | 4e-5 | 4e-5 |
| momentum | 0.9 | 0.9 | 0.9 | 0.9 | —- |
| optimizer | sgd | sgd | sgd | sgd | adam |
| warm-up | True | True | False | False | False |

### A.3  EXPERIMENT RESULT

In this section, we display the change of indicator $\chi$(Fig. 5, Fig. 6), the final $\beta$ and the test accuracy of each experiment. Our codes can be found in the supplemental. As shown by Fig. 7, Fig. 8, Fig. 9 and Fig. 10, except for the first layer of the Sew-ResNet34+ImageNet experiment, almost all neuron layers have obtained a steeper surrogate gradient (a larger $\beta$), which illustrates that surrogate gradient has further optimization space in the SNN training process. However, as we mentioned in Sec. 4.1, choosing a surrogate gradient closer to the $\delta-$function at beginning will make training the network difficult, therefore we'll need tools (such as CPNG) to assist us in finding a better surrogate gradient during training. In the Sew-ResNet34+ImageNet experiment, CPNG finds a smaller $\beta$ for the first layer to ensure the proportion of membrane potential with non-zero gradients. This suggests that when backpropagating with surrogate gradient in deeper networks, the initial few layers may only have a small proportion of their parameters updated, and we should account for this more.

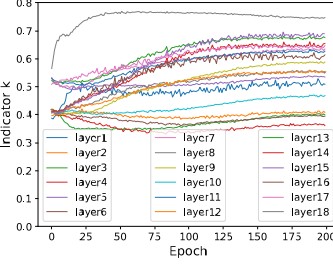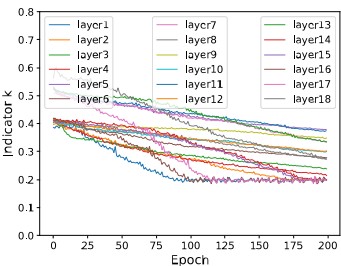

Figure 5: In the experiment of ResNet19, the proportion of non-zero gradient membrane potentials of neurons in different layers without CPNG (left) and with CPNG (right).

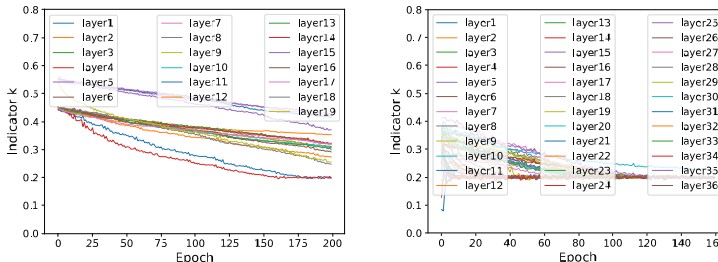

Figure 6: The proportion of non-zero gradient membrane potentials of neurons in different layers with CPNG in the experiment of ResNet18 (left) and Sew-ResNet34 (right).

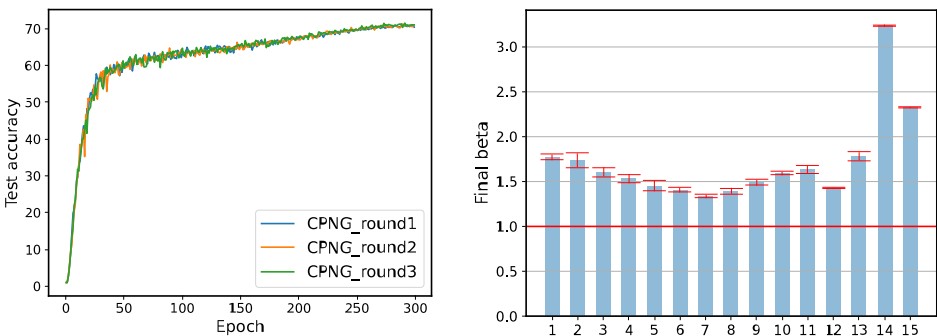

Figure 7: Left: Test accuracy of triplicate VGG16+CIFAR100 experiments using CPNG. Right: Final $\beta$ of each layer in CIFAR100 experiment, the initial $\beta$ is 1.0

## A.4   Using beta obtained from CPNG to train from scratch

We fix each layer's $\beta$ the same as it in Figure 4 (left), the final accuracy is 71.02%, which is 0.3% lower than using CPNG directly and 1.94% higher than the baseline. We keep $\beta = 1$ during the first 1/10 epochs for warm-up, otherwise the accuracy will stay at 1%. This demonstrates that the optimal SG shape discovered by CPNG for all layers is valid, but the result under the case that directly changes the $\beta$ to the optimal value is not as excellent as the case using the CPNG method due to non-stationary transitions.

## A.5   Membrane Potential Distribution

In this section, we show the membrane potential distribution of each layer of neurons in all experiments( Fig. 11, Fig. 12, Fig. 13). The layer after BN is more consistent with the normal distribution than the layer after FC. But for the convenience of calculation, we approximate the membrane potential distribution of all neurons to the normal distribution.

## A.6   Adjust Other Surrogate Gradients

The results we report are based on triangle-like surrogate gradient, a surrogate gradient with finite non-zero gradient interval, but this does not mean that our method can only be used in this case. All results are shown in Tab. 6.

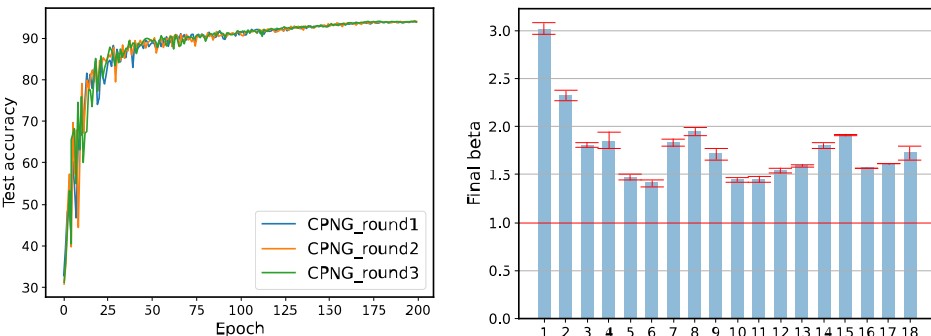

Figure 8: Left: Test accuracy of triplicate ResNet19+CIFAR10 experiments using CPNG. Right: Final $\beta$ of each layer in CIFAR10 experiment, the initial $\beta$ is 1.0

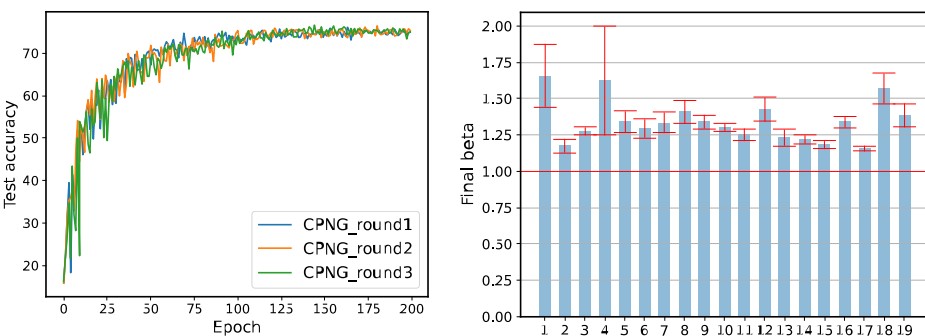

Figure 9: Left: Test accuracy of triplicate ResNet18+CIFAR10-DVS experiments using CPNG. Right: Final $\beta$ of each layer in CIFAR10-DVS experiment, the initial $\beta$ is 1.0

## A.7 SALIENCY MAP

The gradient noise generated by surrogate gradient may affect the network's attention location, resulting in a more blurred saliency map. And there's no doubt that a more appropriate surrogate gradient will produce fewer gradient noises. Thus, the saliency map's clarity may be utilized to determine the fitness of surrogate gradient. We used SmoothGrad (Smilkov et al., 2017), which reduces the effect of visually noise. The results of CIFAR100+VGG16 experiments are shown in Fig. 14. We discovered that CPNG can assist the model to identify clearer contour information. For example, in the boys category, the saliency map obtained by using CPNG is more accurately positioned on the person's face, whereas traditional surrogate gradient also pays much attention to the surrounding background; in the elephants category, the use of CPNG can clearly see the trunk as well as the elephant's outline, whereas traditional surrogate gradient can only obtain blurred borders.

## A.8 LOSS LANDSCAPE

In order to illustrate the trainability of the model after using CPNG, we use the loss landscape to show the change of loss after changing the model parameters. If the loss landscape does not reveal more non-convex regions than the traditional surrogate gradient, it means that using CPNG will not make convergence more difficult. If the same weight offset is applied for various networks when displaying the loss landscape, the networks with bigger weight will exhibit more stationarity. However, due to the existence of batch normalization, the weight scaling of the network has no effect on the inference results, so the sharpness of loss landscape of different networks may only be due to the weight scaling. To explain the model performance between the traditional surrogate gradient and CPNG, we use

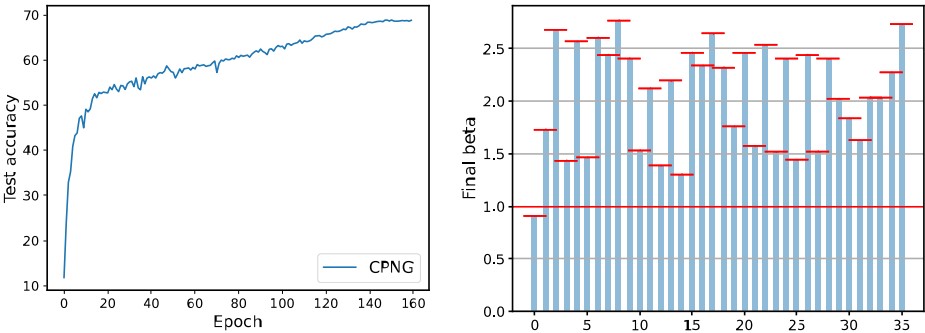

Figure 10: Left: Test accuracy of Sew-ResNet34+ImageNet experiments using CPNG. Right: Final $\beta$ of each layer in ImageNet experiment, the initial $\beta$ is 1.0

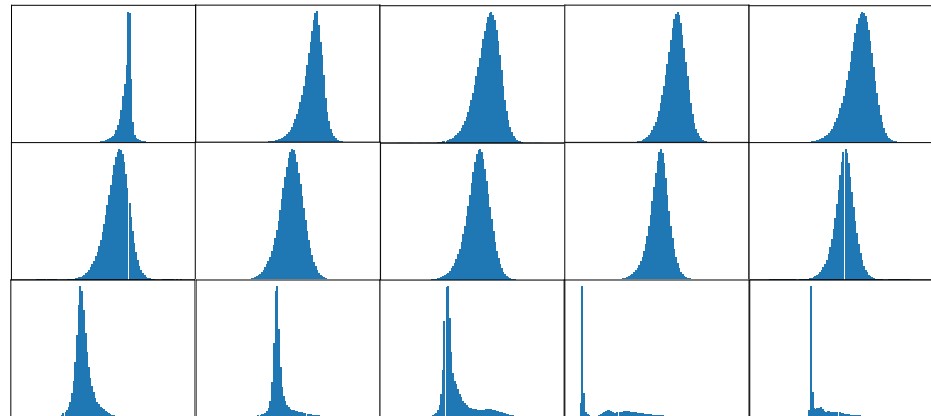

Figure 11: Membrane distribution of each layer in experiment training VGG16 on CIFAR100

the loss landscape demonstration with filter-wise normalization (Li et al., 2017) that mitigates the effect of weight scaling and correlates the model's generalization ability to the flatness of the loss landscape. As shown in Fig. 15, in the experiment of ResNet19+CIFAR10, utilizing CPNG can obtain a smoother minimum and a wider locally convex region, implying that CPNG has no negative effect on network's convergence difficulty.

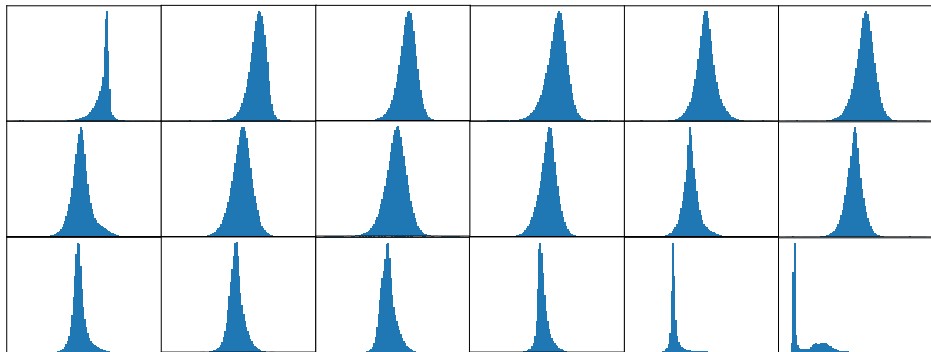

Figure 12: Membrane distribution of each layer in experiment training ResNet19 on CIFAR10

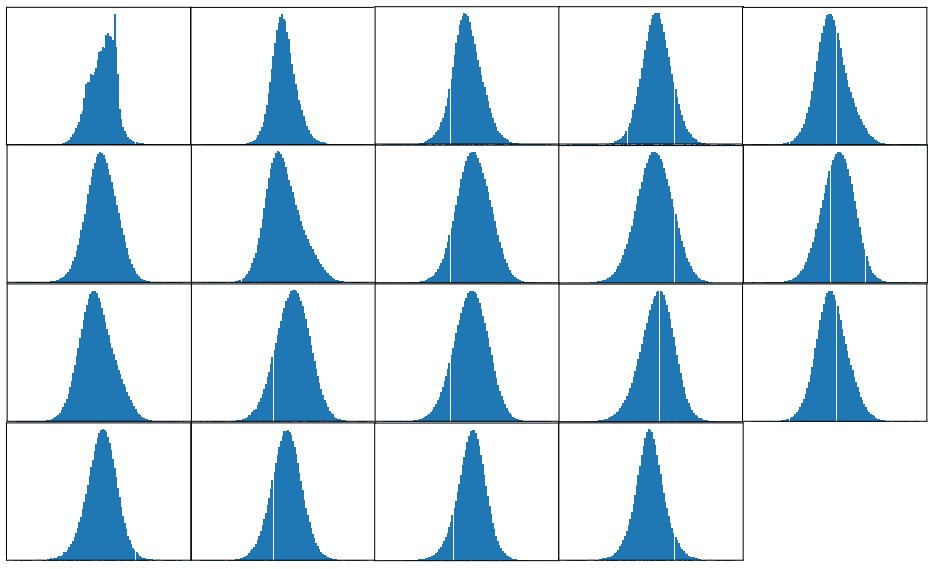

Figure 13: Membrane distribution of each layer in experiment training ResNet18 on CIFAR-DVS

Table 6: Examine CPNG on various surrogate gradients.

| Dataset | Method | Architecture | Time Step | Accuracy |
|---------|--------|--------------|-----------|----------|
| CIFAR10 | Triangular | ResNet19 | 6 | 93.26% |
| | Triangular+CPNG | ResNet19 | 6 | 94.02±0.05% |
| | Rectangular | ResNet19 | 6 | 90.5% |
| | Rectangular+CPNG | ResNet19 | 6 | 93.07±0.04% |
| | ArcTan | ResNet19 | 6 | 93.24% |
| | ArcTan+CPNG | ResNet19 | 6 | 93.95% |
| CIFAR100 | Triangular | VGG16 | 5 | 69.08% |
| | Triangular+CPNG | VGG16 | 5 | 71.32±0.20% |
| | Rectangular | VGG16 | 5 | 65.54% |
| | Rectangular+CPNG | VGG16 | 5 | 67.74±1.02% |
| | ArcTan | VGG16 | 5 | 68.15% |
| | ArcTan+CPNG | VGG16 | 5 | 69.49±0.20% |
| | Triangular+CPNG | ResNet20 | 6 | 76.09% |
| CIFAR10-DVS | ArcTan | ResNet18 | 10 | 67.2% |
| | ArcTan+CPNG | ResNet18 | 10 | 67.5% |
| | Triangular+TET | ResNet18 | 10 | 79.9% |
| | Triangular+TET+CPNG | ResNet18 | 10 | 82.3% |
| | Triangular | ResNet18 | 10 | 75.6% |
| | Triangular+CPNG | ResNet18 | 10 | 76.37±1.02 % |

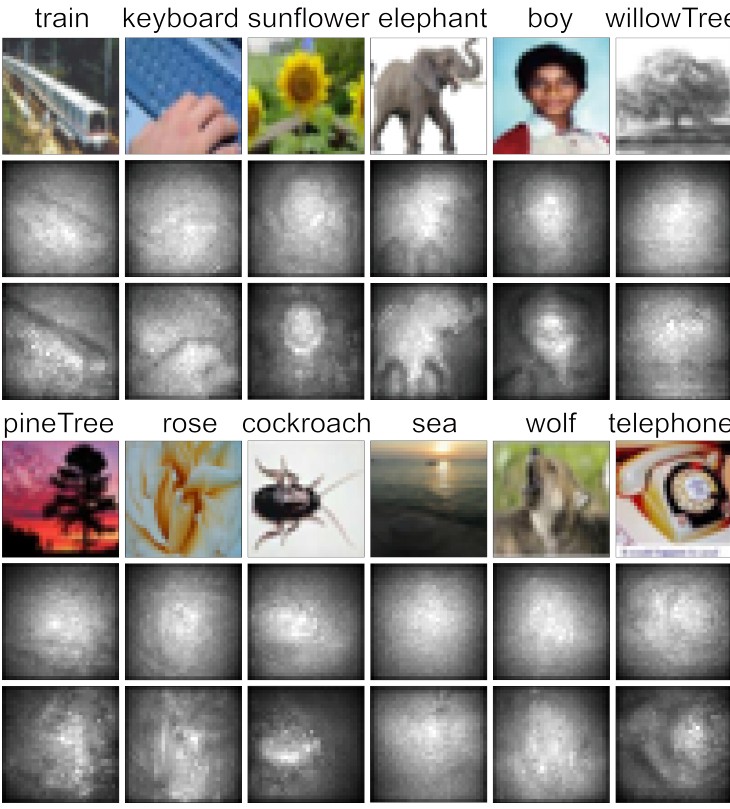

Figure 14: Saliency map. Three photos are a group, the top is the original image, the middle is the salilency map given by the model trained by traditional surrogate gradient, and the bottom is the saliency map given by the model trained with CPNG.

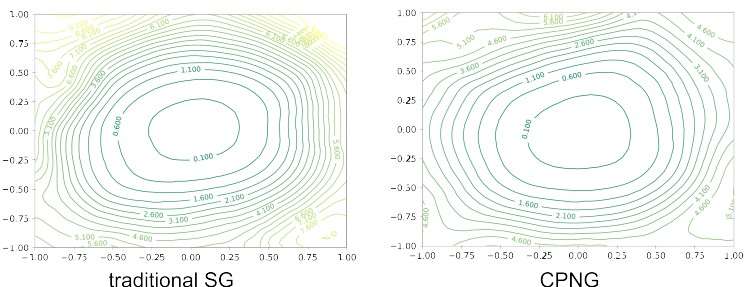

Figure 15: Loss landscape of traditional surrogate gradient and CPNG.

