# OpenReview forum: "Efficient Surrogate Gradients for Training Spiking Neural Networks"
_ICLR.cc/2023/Conference — Submitted to ICLR 2023_

### Official Review · Reviewer_eFt1 · 2022-10-18

**Confidence:** 5
**Correctness:** 3
**Technical Novelty And Significance:** 2
**Empirical Novelty And Significance:** Not applicable
**Recommendation:** 3

**Clarity, Quality, Novelty And Reproducibility:**

Paper is very clearly written. Very limited novelty. I believe the results can be reproduced.

**Details Of Ethics Concerns:**

I do not see any ethical concerns

**Strength And Weaknesses:**

Strength: The idea introduced is intuitive, experiments show improvements.
Weekness: The idea is fairly obvious. Implementation is rather laborious and there are no formal/mathematical evaluations of why this intuitive solution ahould give better performance. The authors just suggest a technique and then evaluate it on a set of datasets.

**Summary Of The Paper:**

This paper is about changes that need to be made to error-backpropagation in order to train a leaky-integrate and fire spiking neuron network. A technique that is commonly used is to "smoothen" the spike so that gradients do not collapse onto the two values of 0 or infinity. This paper suggests an adaptive smoother. They define a parametrized class of continuous functions (triangles with compact support in their case) that limit to the delta function. They then adapt the support parameter (the width of the triangle) based on the layer and the input. Specifically, (based on samples), they determine what the range of values the membrane potential takes and then use that to set the width parameter to have non-zero finite gradients for a fixed proportion of neurons.

**Summary Of The Review:**

The introduction of "adaptation" to the surrogate gradient is not novel enough to warrant publication, particularly since there have been so many ways people have tried to get around the gradient issue for spiking neuron networks.

---

> ### Author Response · Authors · 2022-11-14
> **Response to Reviewer eFt1**
>
> ### Comment 1:
> The idea is fairly obvious. Implementation is rather laborious and there are no formal/mathematical evaluations of why this intuitive solution should give better performance. The authors just suggest a technique and then evaluate it on a set of datasets. The introduction of "adaptation" to the surrogate gradient is not novel enough to warrant publication, particularly since there have been so many ways people have tried to get around the gradient issue for spiking neuron networks.\\
> ### Response to comment 1:
> The major contribution of the current work is to identify the problem of effective domain in applying a surrogate gradient to train spiking neural networks. We propose a pipeline to control the non-zero gradients (CPNG) that applies to many types of surrogate gradients, including ArcTan, triangle, and rectangle, and consistently improves the accuracy. Our work is founded on straightforward concept of finding a better surrogate gradient that means to balance the surrogate gradient feasible region and its similarity to the actual gradient ($\delta$-function). It is true that even the feasibility of the surrogate gradient itself lacks theoretical proof. However, compared to empirically adjusting the surrogate gradient shape, our method based on the domain efficiency of surrogate gradients is more reliable. In addition, we employ the saliency map (appendix 6) and loss landscape (appendix 7) to demonstrate that the CPNG has reduced gradient noise and that the trained SNN has superior generalization. Besides, we disagree with your opinion that the surrogate gradient is irrelevant. Alternative approaches to obtain deep SNNs, such as ANN-to-SNN conversion, cannot design the network structure as arbitrarily as direct training, cannot use very low time steps (e.g., 2, 4), and cannot handle neuromorphic datasets. As a required module for SNN direct training, the design of the surrogate gradient is important.

---

> ### Comment · Area_Chair_vLoN · 2022-11-20
> **Please respond to author rebuttal**
>
> Dear Paper791 Reviewer eFt1,
> I wonder whether you could respond to the author rebuttal? Many thanks!
>
> AC

---

### Official Review · Reviewer_Sqv2 · 2022-10-21

**Confidence:** 4
**Correctness:** 3
**Technical Novelty And Significance:** 3
**Empirical Novelty And Significance:** 3
**Recommendation:** 6

**Clarity, Quality, Novelty And Reproducibility:**

There are implementation details which are not clear to me and make me feel like algorithm 1 is unnecessarily complicated, but this might be a matter of clarifying what has been implemented and tested. It is probably related to the performance in Table 1 which I could not understand because the algorithm CNPG(B) and CNPG(E) are not defined precisely.

My understanding is that CNPG can be implemented easily with the current batch without keeping track of $\mathcal{X}_{record}$. This simple version would be: before the applying the gradient, the vectors of $u$ is available and could be used to compute the corresponding $\mathcal{X}$ and therefore $\beta$ directly. Maybe this was version (B), if yes this should be clarified. Then version (E), if it corresponds to algorithm 1, seems to be almost identical to perform the same calculation for $\beta$ but to apply some kind of moving average on it. If this simple explanation is a good summary for Algorithm I think it would help the reader to make it clear.

**Strength And Weaknesses:**

The method is simple and it can be implemented easily with auto-diff which is a strong plus. The benefit seems to be substantial although more details are required to make sure the benefit comes from the adaptive scaling keeping a stable proportion of zeros and not a naive scaling of beta. So I would find the results much stronger if the performance difference between triangular and triangular+CNPG is investigated further to understand better how it works, I suggest a couple of clarifications and simple experiments which would make the paper better in my opinion:

1) What is the value of beta for the Triangular control on the performance table? Traditionally the performance of the baseline would be optimized by trying different values of beta, for instance in [02, 0.5, 0.8, 1, 1.5, 2., 3.], was it done in the performance table? If not it would make sense to try, or to report at least how beta was chosen and add the performance obtained with the average beta value for Figure 4 (beta=1.5 looks like a good control for VGG16 for instance).

2) Given that the method is rather ad-hoc it is rather important to demonstrate what aspects of the method matters the most. For instance, I wonder whether restarting a training with fixed layer specific betas as shown in Figure 4 would also explained the performance benefit. Crucially, even if this control is good, it does not mean that CNPG is useless because it finds those beta values automatically, but at least it would provide some insights on whether the adaptive aspect of the method is important or whether what matters is to find the right betas.

A strong weakness is that this paper seems embedded into a sub-community of SNN research and ignores many relevant work outside of that:

3) All of what is applied here for surrogate gradient is equally applicable to binary network where the relu activation function is simply replaced by a Heaviside function. A connection to this literature and a fair account of it's progress would be relevant, see [1] for the first publication on back-prop with quantized activation and [2] for the first paper using the triangular derivative as far as I know. On top of citing these two papers, I would find the paper much stronger if a binary ANN model like Esser et al. (basically equivalent to SNN with T=1) would be added to show that the method can also be applied in the broader quantized network community. Along those line, providing the performance of the model with relu and all identical parameters would be a good insight to see how far is the model compared to classical ANNs.

4) There is already an open debate inside the SNN community about the benefit of triangular derivatives and how their sharpness was identified to be a critical parameter, see methods from [3] for instance where the height of beta is referred to as the dampening factor. It would be fair to acknowledge that.

[1] Estimating or Propagating Gradients Through Stochastic Neurons for Conditional Computation
Yoshua Bengio, Nicholas Léonard, Aaron Courville

[2] Backpropagation for Energy-Efficient Neuromorphic Computing
Steve K. Esser, Rathinakumar Appuswamy, Paul Merolla, John V. Arthur, Dharmendra S. Modha

[3] Long short-term memory and learning-to-learn in networks of spiking neurons
Guillaume Bellec, Darjan Salaj, Anand Subramoney, Robert Legenstein, Wolfgang Maass

**Summary Of The Paper:**

A specific surrogate gradient is described, the idea is to regulate the sharpness of the triangular surrogate gradient at each iteration so that the proportion of non-zero gradients is controlled at each layer. This simple method can be implemented efficiently and leads to a ~2% improvement on ImageNet and CIFAR 100.

**Summary Of The Review:**

Overall the method is simple and might be an important step towards binary ANN and SNN, my hope is that a method of this type can make it perform as-well as real-valued Relu networks. If the claim is true it is one step in that direction. This could have a tremendous impact but a few clarifications are necessary to make sure that the performance gain is coming from the method as described in the paper.

-- Review update, November 6th --
I was eager to try so I implemented a simple version of the algorithm in this paper. I tried this approach on a Binary MLP on MNIST with 2 hidden layers (no time steps). On this simple setup the performance was lower than with the regular triangular derivative with constant width. My implementation was certainly not identical, so there might be a way to make it work. I think this paper would have been stronger if it described a simple reproducible setup of this kind.

---

> ### Author Response · Authors · 2022-11-14
> **Response to Reviewer Sqv2 (1)**
>
> Thank you for reviewing our paper and providing valuable suggestions. We will clarify our contribution and technical novelty below in the response to your questions. Hope it can relieve your concerns and make our work stronger from your perspective.
>
> ### Comment 1:
> What is the value of beta for the Triangular control on the performance table? Traditionally the performance of the baseline would be optimized by trying different values of beta, for instance in [02, 0.5, 0.8, 1, 1.5, 2., 3.], was it done in the performance table? If not it would make sense to try, or to report at least how beta was chosen and add the performance obtained with the average beta value for Figure 4 (beta=1.5 looks like a good control for VGG16 for instance).
> ### Response to comment 1:
> We use $\beta=1$ as the initial value in all experiments, and $\beta=1$ is the default setting in the previous works [1, 2]. We have tried different $\beta$ in VGG16 experiments (e.g., 0.25, 0.5, 1, 1.5, 2.0 in Figure 1) and the experimental results show that $\beta=1$ is a good setting. However, due to computational costs, we did not perform hyperparametric searches for all experiments. We will give a more advanced hyperparametric search in our final version. In Figure 4, the final average beta obtained by CPNG is around 1.5. But, according to the experimental results in Figure 1 for $\beta=1.5$, directly using $\beta=1.5$ from scratch will result in training failure.
>
> ### Comment 2:
> Given that the method is rather ad-hoc it is rather important to demonstrate what aspects of the method matter the most. For instance, I wonder whether restarting a training with fixed layer-specific betas as shown in Figure 4 would also explain the performance benefit. Crucially, even if this control is good, it does not mean that CNPG is useless because it finds those beta values automatically, but at least it would provide some insights into whether the adaptive aspect of the method is important or whether what matters is to find the right betas.
>
> ### Response to comment 2:
> Thanks for your suggestion, we think it is a meaningful experiment, and we have added this experiment in the appendix. On VGG16, we fix each layer's $\beta$ the same as it in Figure 4, the final accuracy is 71.02\%, which is 0.3\% lower than using CPNG directly and 1.94\% higher than the baseline. We keep $\beta=1$ during the first 1/10 epochs for warm-up, otherwise, the accuracy will stay at 1\%. This demonstrates that the optimal SG shape discovered by CPNG for all layers is valid, but the result under the case that directly changes the $\beta$ to the optimal value is not as excellent as the case using the CPNG method due to non-stationary transitions.
>
> ### Comment 3:
> A strong weakness is that this paper seems embedded into a sub-community of SNN research and ignores many relevant works outside of that (binary network).
> ### Response to comment 3:
> In fact, the EDE method proposed in IR-Net [3] also tries to progressively adjust the surrogate gradient to approximate the $\delta-$function during the training phase, and the efficacy of EDE has been verified in their ablation study, which shows EDE enhances BNN accuracy by 1.4\%. In comparison to EDE, which uses the same surrogate gradient for all layers and adjusts the surrogate gradient shape according to the training epoch number, we believe that the CPNG method is more reasonable. We use the IR-Net source code, replace EDE with CPNG, and set the initial values of k and t to 1. We get results of 84.34\% (w/o CPNG) and 86.25\%(with CPNG), respectively, while the result we got from training with the source code (w/ EDE) was 85.76\% (random seed 1028, the reported accuracy is 85.4\%).
>
>
> ### Comment 4:
> There is already an open debate inside the SNN community about the benefit of triangular derivatives and how their sharpness was identified to be a critical parameter, see methods from [3] for instance where the height of beta is referred to as the dampening factor. It would be fair to acknowledge that.
> ### Response to comment 4:
> Thanks for the reference, we will discuss them in the related work section. In our work, the integral of the surrogate gradient is maintained at 1 in accordance with the $\delta$-function. Therefore, the surrogate gradient does not generate mistakes on the integral, and we can adjust the feasible region by changing $\beta$. Using CPNG with varied damping factors is very interesting, but it adds a significant amount of work and is outside the scope of the paper's current investigation, which is unable to be completed during the rebuttal period. We will add these experiments to the final version.

---

> > ### Author Response · Authors · 2022-11-14
> > **Response to Reviewer Sqv2 (2)**
> >
> > ### Comment 5:
> > There are implementation details which are not clear to me and make me feel like algorithm 1 is unnecessarily complicated, but this might be a matter of clarifying what has been implemented and tested. It is probably related to the performance in Table 1 which I could not understand because the algorithm CNPG(B) and CNPG(E) are not defined precisely. My understanding is that CNPG can be implemented easily with the current batch without keeping track of ${\chi}_\text{recorder}$. This simple version would be: before the applying the gradient, the vectors of $\mu$ is available and could be used to compute the corresponding $\chi$ and therefore $\beta$ directly.
> >
> > ### Response to comment 5:
> > CNPG (B) means adjusting the surrogate gradient's shape per batch, and CNPG (E) means adjusting the surrogate gradient's shape per epoch. We have explicitly defined them in the paper. In any case, we require a space to store $\chi_{target}$. In our code, we save it using $\chi_{recorder}$. The solution you suggested might be to save each layer's $\chi_{target}$ in a different space (such as the LIF layer). According to Table 1, CPNG (E) provides a greater performance improvement with a smaller time cost than CPNG (B). When using CPNG, we expect that the network parameters are appropriate, that is, the network has traversed the whole dataset to prevent the misleading of network parameters by data randomness.  For example, if we use CPNG once per batch, the network parameters are mostly affected by the first few batches in the early stages of network training, and the statistical indicator $\chi_{recorder}$ may fast converge to $\chi_{limit}$.
> >
> > ### Comment 6:
> > I was eager to try so I implemented a simple version of the algorithm in this paper. I tried this approach on a Binary MLP on MNIST with 2 hidden layers (no time steps). On this simple setup the performance was lower than with the regular triangular derivative with constant width. My implementation was certainly not identical, so there might be a way to make it work. I think this paper would have been stronger if it described a simple reproducible setup of this kind.
> > ### Response to comment 6:
> > We have done some experiments (T=1) according to the situation you encountered, and the results are as follows, our experiments show that CPNG is still effective when $T=1$.
> > | Dataset | Architecture | CPNG | Accuracy|
> > |  ----  | ----  |  ----  | ----  |
> > | MNIST | Conv64-Conv128-Pool-FC10 (Vth=1.0)} | False | 98.54\%
> > | MNIST | Conv64-Conv128-Pool-FC10 (Vth=1.0)} | True | 98.60\%
> > |MNIST  | FC784-FC10 (Vth=1.0)}                           | False | 98.33\%
> > |MNIST  | FC784-FC10 (Vth=1.0)}                           | True  | 98.36\%
> > |CIFAR100 | ResNet18 (Vth=0.5)                 | False | 68.24\%
> > |CIFAR100 | ResNet18 (Vth=0.5)                 | True   | 69.32\%
> >
> > We guess that what you are encountering may be due to the fact that you are implementing CPNG(B), using  CPNG(E) or warm-up may lead to higher stability of the algorithm. In general, CPNG is more effective on deeper networks, where the mismatch of gradients is more severe.
> >
> >
> > [1] Rathi N, Roy K. Diet-snn: Direct input encoding with leakage and threshold optimization in deep spiking neural networks[J]. arXiv preprint arXiv:2008.03658, 2020.
> >
> > [2] Deng S, Li Y, Zhang S, et al. Temporal Efficient Training of Spiking Neural Network via Gradient Re-weighting[C]//International Conference on Learning Representations. 2021.
> >
> > [3] Qin H, Gong R, Liu X, et al. Forward and backward information retention for accurate binary neural networks[C]//Proceedings of the IEEE/CVF conference on computer vision and pattern recognition. 2020: 2250-2259.

---

> > > ### Comment · Reviewer_Sqv2 · 2022-11-22
> > > **Thank you**
> > >
> > > Thank you for the great rebuttal! And thank you for addressing seriously my critics and suggestions.
> > >
> > > The authors have convinced me that the method is a progress towards better surrogate gradients, hence it is worth publication.
> > > In my MNIST I just used a fully connected layer with 2 hidden layers with (CNPG-B), the method is indeed slightly better than the regular surrogate gradient although but it is still significantly worse than the same relu baseline.
> > >
> > > I am increasing my grade to 6 now, assuming that the authors will do the following:
> > > - Report that their method also provides a small improvement when T=1 which is standard binary network setting. It shows that the work might be impactful beyond the spiking network community.
> > > - Fairly acknowledge the prior work on the optimization of triangular derivatives
> > > - Specify what is the network architecture and dataset used in Figure 1
> > >
> > > I believe authors have the opportunity to modify the draft. I would encourage the authors to do so, so that we can evaluate as closely as possible what will be published.

---

> ### Comment · Area_Chair_vLoN · 2022-11-20
> **Please respond to author rebuttal**
>
> Dear Paper791 Reviewer Sqv2,
> I wonder whether you could respond to the author rebuttal? Many thanks!
>
> AC

---

### Official Review · Reviewer_yRYU · 2022-10-23

**Confidence:** 3
**Correctness:** 2
**Technical Novelty And Significance:** 4
**Empirical Novelty And Significance:** 4
**Recommendation:** 6

**Clarity, Quality, Novelty And Reproducibility:**

This reviewer has an ambivalent opinion about the clarity aspect of this paper. On one hand, this paper demands a lot of background in the field, using a lot of field-specific jargon, on the other hand, as soon as the key concepts are established, it is rather easy to follow the main narrative of the paper. Formulas are very thoroughly explained, which helps to understand the intuition behind the proposed method.

The proposed method to control the proportion of non-zero gradients seems to be a novel contribution to this work. For this reviewer who is not an expert in the Spiking Neural Networks field, after a brief scan of related literature (including detailed analysis in this paper) this seems to be a significant contribution to the field.

This reviewer is the least confident in the quality evaluation, because of the issue described in the previous section (i.e. hiding the performance of the alternative approaches). Leaving it aside, the technical depth and experimental design of this publication seem reasonable. There is enough evidence gathered by the authors of the paper to support every claim. But the issue highlighted before seems to be rather substantial, lowering the overall recommendation.

Finally, this work appears to adhere to high standards of reproducibility: the source code in Python is provided, as well as the pseudo algorithm of the proposed approach is published. All the datasets used in the paper are publicly available, hence it is possible to re-run all the described experiments and confirm the results.



**Details Of Ethics Concerns:**

As pointed out above, this seemed as if when compared to other methods, the authors picked easier comparators, while avoiding those that might make their approach seem inferior. This reviewer is not certain if this is substantial ground for the ethics review, but it seemed important. Also, the reviewer admits the lack of relevant experience in the field, hence admits that the probability of a mistake is high.

Update on 25.11: the authors have addressed my concerned above, promising to clarify the comparison with existing approaches, hence I increase my ranking from 3 to 6.

**Strength And Weaknesses:**

This paper seems to have a substantial technical depth, overall, for the non-expert, the method proposed in this work seems to be very reasonable and novel. If not for the worrying weakness, presented below, this reviewer would argue for this paper to be accepted.

According to this reviewer, the main weakness of this paper is in the results, more precisely in the section Comparison to Existing Work. Going through Table 4 the reviewer could not help to notice some inconsistencies. For example, when discussing the results on CIFAR100 with VGG16 the authors wrote that "CPNG manages to improve 2.24% accuracy", while it remained unclear for the reviewer from Table 4, where this 2.24% is coming from. Diet-SNN with an accuracy of 69.67% is 1.65% below the performance of CPNG (71.32%). Also perhaps more surprisingly the other two methods presented in Table 4, under the same CIFAR100 category show much better accuracies, ranging from 73.55% (ANN-to-SNN with time step 32 by Li et al.) all the way to 77.40% (ANN-to-SNN with time step 128 by Li et al.). Another important observation is that the results by Dspike (Li et al., 2021b) are not presented in this category at all, despite the fact that the Dspike performs relatively well on CIFAR100, with superior to CPNG performance of 74.24%. The same or similar situation is with other datasets leaving only CIFAR10-DVS, Dspike is left out of the comparison with usually superior performance. All these create an impression that the authors were trying to manipulate the readers by hiding information that presents their method in an unfavorable light. This reviewer is happy to be proven wrong and pointed at a different reason for these inconsistencies e.g. because Dspike had used a different part of the validation or test set hence making the results incomparable. Nevertheless, even if these other reasons exist for not presenting the results from Dspike, this reviewer strongly believes that the readers should have been surely presented with these reasons in advance. Showing less favorable performance compared to some alternatives should not encourage the authors to hide these alternatives, but rather admit that some other methods seem to have an edge.

Generally, a paper does require a lot of relevant background, hence might be very hard to read for non-experts. Terms such as delta-function, membrane potential neurons, pre-synaptic input, leaky integrate and fire module, and others are used without being defined nor explained, raising the bar for researchers outside of the spiking neural networks field. Having said this, formulas proposed in the paper to illustrate some key concepts are thoroughly defined and explained, making it possible to follow the narrative albeit missing some details. Another formatting issue that might become apparent for researchers outside of the field who are not as well familiar with terminology is the use of acronyms that were either not defined or defined. I.e. one time defined acronym somewhere in the abstract and later used arbitrarily in the text, is going to be confusing for the unprepared reader e.g. "SG", "tdBN", etc.

**Summary Of The Paper:**

Spiking Neural Networks (SNN) are biologically more plausible, capable of representing time as well as more energy-efficient versions of more common artificial neural networks. The main difference is that SNNs communicate across layers using additive spikes. However besides the aforementioned advantages, this makes it really hard to train such networks. Three alternatives exist to do it. One is to convert pre-trained ANN to SNN, use heuristics, and lastly use a type of backpropagation method to train these networks. Surrogate gradients help to train SNNs. However, even using surrogate gradients SNNs train less efficiently. Also, the resulting performance of SNNs is usually worse than the one of ANNs.This is because surrogate gradients can only approximate real gradients, this procedure is very susceptible to the problem of vanishing gradients and is overall less stable than in ANNs. The main idea behind this paper is to stabilize the process of training SNNs by keeping track of the proportion of non-zero gradients during training. Experimental results presented in the paper seem to suggest that the proposed strategy produces superior results to published alternatives.

**Summary Of The Review:**

Overall, the paper leaves a very good impression from the novelty and reproducibility points of view. Some additional work should be done to improve the clarity of the paper, especially for people who lack the SNN background. Most importantly, additional work is expected to explain the inconsistencies pointed out in the results when it comes to comparison to the existing methods. At the moment it seems as if the authors avoid displaying information that might make their method look inferior. In goodwill, this reviewer leaves enough space for the authors to prove the reviewer wrong, therefore there is a good chance that my recommendation will change as soon as evidence to the contrary is presented.

---

> ### Author Response · Authors · 2022-11-14
> **Response to Reviewer yRYU**
>
> Thank you for reviewing our paper and raising your confusion here. We are happy to clarify and discuss these points. First, regarding the experiments, we are more than familiar with the works you mentioned above, and we never attempt to manipulate the comparators to make our results appear strong. The main reason is that those methods applied additional processing steps or paradigms which are not included in other comparators. We will explain this in detail and provide additional results below. If we align these conditions, our method still outperforms them.  As serious researchers that release codes for all our published machine learning papers in many conferences, we take academic ethics as our bottom line thus we won't accept this accusation. Hope our response below can address your concerns on this point.
>
>
> Regarding the clarity of statements, thank you for suggesting adding more definitions of fundamental terms. We will add this to our manuscript to make it readable to a broader group of researchers.
>
> ### Comment 1:
> When discussing the results on CIFAR100 with VGG16 the authors wrote that "CPNG manages to improve 2.24\% accuracy", while it remained unclear for the reviewer from Table 4, where this 2.24\% is coming from. Diet-SNN with an accuracy of 69.67\% is 1.65\% below the performance of CPNG (71.32\%).
> ### Response to comment 1:
> We apologize for the confusion. Due to space limitations, we have moved a portion of the experimental results without CPNG to the appendix. Please refer to Table 6 for our experimental results of 69.08\% (w/o CPNG) and 71.32\% (with CPNG).
>
>
> ### Comment 2:
> Also perhaps more surprisingly the other two methods presented in Table 4, under the same CIFAR100 category show much better accuracies, ranging from 73.55\% (ANN-to-SNN with time step 32 by Li et al.) all the way to 77.40\% (ANN-to-SNN with time step 128 by Li et al.).
> ### Response to comment 2:
> All two methods are conversion-based. They first train a high-performing ANN network (Li et al. 77.89\%, Bu et al. 76.28\%) and then convert it to SNN, making the SNN every layer's spike frequency close to the activation of the source ANN. The SNN obtained by the conversion-based method usually has poorer results than direct training when T is very small. According to Table S4 by Bu et al., the accuracy is 63.79\% when $T=2$ and 69.62\% when $T=4$. While CPNG has achieved an accuracy of 66.14\% (IF, $V_{th}=0.5$) when $T=2$ and an accuracy of 69.83\% when $T=4$.
>
> ### Comment 3:
> Another important observation is that the results by Dspike (Li et al., 2021b) are not presented in this category at all, despite the fact that the Dspike performs relatively well on CIFAR100, with superior to CPNG performance of 74.24\%.
> ### Response to comment 3:
> Dspike uses a different network structure in CIFAR100, which uses 3 stem layers and avg-downsample block and similar to ResNet-20 architecture in our works (mentioned in Section 5.1 of the Dspike paper). Besides, Li et al. use a well-trained ANN to initialize parameters and train the network with their TIT technique. Using the same structure SNN on CIFAR100 as Li et al., we obtain an accuracy of 76.09\% when training from scratch with CPNG. This performance is 1.85\% higher than the previous work. We have added the results to the rebuttal version.
>
> ### Comment 4:
> Another formatting issue that might become apparent for researchers outside of the field who are not as well familiar with terminology is the use of acronyms that were either not defined or defined.
> ### Response to comment 4:
> Thank you for your suggestion; we will check the article to fix these issues.

---

> > ### Comment · Reviewer_yRYU · 2022-11-22
> > **Respond to author rebuttal**
> >
> > Dear authors,
> >
> > thank you for addressing my concerns. Just a comment about Dspike, I believe that in the original paper they have also presented results for CIFAR10 and ImageNet. In both these cases, they again show better performance albeit using slightly different architectures. Have you checked what would be CPNG's performance using the same architectures?
> >
> > In general, I believe that you could have done more early on to leave no room for such ambiguities.

---

> > > ### Author Response · Authors · 2022-11-22
> > > **Response to Reviewer yRYU**
> > >
> > > Thank you for your response. Dspike is valid but has a lot of overhead, so we present a technique (CPNG) for the low-overhead search of the surrogate gradient. Each time Dspike adjusts the surrogate gradient, it needs to perform many forward propagations (if the weight shape of the first convolutional layer of the network is $[dim_{out}, dim_{in}, kernelSize[0], kernelSize[1]]$, you need to perform $3 * dim_{out} * dim_{in} * kernelSize[0] * kernelSize[1]$ forward propagations. You may find that Dspike takes $85$ seconds to train CIFAR-DVS for one epoch while tuning surrogate gradient takes an additional $280$ seconds), which makes Dspike unable to perform many tasks. Compared with Dspike, our method is fast and can be better combined with other methods (such as TET). The CIFAR10 and CIFAR100 experiments are intended to demonstrate that CPNG can be used in various network structures (VGG, ResNet), rather than pursuing the SOTA. The improvement brought about by CPNG, not the final accuracy, is what we want to demonstrate most. In experiments on CIFAR-DVS, we show that combining CPNG with TET can achieve new SOTA.

---

> > > > ### Comment · Reviewer_yRYU · 2022-11-25
> > > > **Thank you for clarification**
> > > >
> > > > Dear authors,
> > > >
> > > > thank you for addressing my comments and answering my questions.
> > > >
> > > > I believe that it is possible to add a lot more clarity to the comparison with the existing methods and with Dspike in particular. For example, one could add training time and inference time to Table 4 to show that although the accuracy of some other approaches is marginally higher in some datasets, they are a lot less practical than the proposed CPNG method.
> > > >
> > > > Provided that the authors improve this aspect, I am ready to raise my ranking to 6.

---

> ### Comment · Area_Chair_vLoN · 2022-11-20
> **Please respond to author rebuttal**
>
> Dear Paper791 Reviewer yRYU,
> I wonder whether you could respond to the author rebuttal? Many thanks!
>
> AC

---

### Official Review · Reviewer_z3Wm · 2022-10-24

**Confidence:** 4
**Clarity, Quality, Novelty And Reproducibility:** The work is clear enough and appears …
**Correctness:** 3
**Technical Novelty And Significance:** 4
**Empirical Novelty And Significance:** Not applicable
**Recommendation:** 6

**Strength And Weaknesses:**

Strengths

The paper contributes to improving the surrogate gradient training of SNNs, which should be practical and profitable for the related field of research. The method is relatively simple and well explained to be reproduced.
The computational costs of the method are properly indicated and remain limited.

Weaknesses

When comparing with the baseline (i.e. without CPNG), it may not be clear that the value of the fixed width is optimised. For a fairer comparison, a hyperparameter search could be done, so that CPNG is compared to the best version of the fixed surrogate gradient function.
In relation to that, in table 3 and 6, confidence intervals are only given for CPNG entries. It would be coherent to also include such intervals for the non-CPNG baseline.
It appears that the choice of value for the effective domain indicator is only based on a single ad-hoc experiment with two values considered, namely 0.05 and 0.2. Here It would also be more meaningful to make a more advanced hyperparameter search.
The grammar could be improved.

Remarks

Given that membrane potentials evolve over time, it is not exactly clear to me how the effective domain indicator is calculated. I can guess that the mean “mu” is computed over all time steps and all neurons, but it may be clearer to indicate it more explicitly inside the text.
It could be interesting to relate the effective domain indicator to the resulting firing rates. Does increasing the former have an impact on the latter?
The method is also only tested using a relatively small number of time steps (5-10). It would be good either to increase the number of steps on the same tasks, or to directly apply it to tasks that involve longer sequences such as speech for instance.

**Summary Of The Paper:**

This paper proposes a method to improve the training of spiking neural networks (SNNs) via the surrogate gradient method. Their technique called CPNG controls the proportion of non-zero gradients during network backpropagation. Surrogate gradient functions are only non-zero when the membrane potential is within a certain region about the threshold value. The width of that region can be defined using a single parameter, which has so far remained fixed during training. In order to allow this parameter to vary optimally during training, the authors compute the proportion of neurons that have their membrane potential within that region of non-zero gradients, and adapt the region accordingly. If the proportion is high enough, the width decreases, and inversely, if the proportion is too small, the width increases, so that the overall activity of the network neither explodes nor vanishes. Their method shows performance improvements on all four tasks (CIFAR10, CIFAR100, CIFAR10-DVS and ImageNet) compared to the fixed width surrogate gradient.



**Summary Of The Review:**

The authors have the right approach; the method makes sense and is helpful.  However, the evaluation could be improved.

---

> ### Author Response · Authors · 2022-11-14
> **Response to Reviewer z3Wm**
>
> Thank you for providing insightful comments and suggestions. Regarding your concerns, we respond piece by piece below and hope our explanation clarifies the concerns here. In terms of your concerns, we provide some additional results here and will include a more complete version in the revised manuscript.
>
>
> ### Comment 1:
> When comparing with the baseline (i.e. without CPNG), it may not be clear that the value of the fixed width is optimized. For a fairer comparison, a hyperparameter search could be done, so that CPNG is compared to the best version of the fixed surrogate gradient function.
>
> ### Response to comment 1:
> Thanks for your useful suggestion, Fig. 1 can be regarded as a preliminary search work, which shows that $\beta=1$ is a good setting. The more advanced optimal hyperparameter search cannot be finished during the rebuttal phase because of its enormous burden. We will provide the search results in our final version.
>
> ### Comment 2:
> In table 3 and 6, confidence intervals are only given for CPNG entries. It would be coherent to also include such intervals for the non-CPNG baseline.
> ### Response to comment 2:
> We will add the non-CPNG intervals in the final version.
>
>
> ### Comment 3:
> It appears that the choice of value for the effective domain indicator is only based on a single ad-hoc experiment with two values considered, namely 0.05 and 0.2. Here It would also be more meaningful to make a more advanced hyperparameter search.
> ### Response to comment 3:
> Table 3 gives our search results for $\chi_{\text{limit}}$. The final accuracy is insensitive to changes in $\chi_{\text{limit}}$ when $\chi_{\text{limit}}$ is less than or equal to 0.3. We will provide more advanced search results in our final version.
>
>
> ### Comment 4:
> The grammar could be improved.
> ### Response to comment 4:
> We have rechecked the grammar.
>
>
> ### Comment 5:
> Given that membrane potentials evolve over time, it is not exactly clear to me how the effective domain indicator is calculated.
> ### Response to comment 5:
> The $\chi$ is calculated at the last batch of each epoch. For each layer, we record the membrane potential of all neurons in every time step (a tensor shaped like batchsize-by-timestep-by-channels-by-H-by-W) and calculate the mean and variance.
>
> ### Comment 6:
> It could be interesting to relate the effective domain indicator to the resulting firing rates. Does increasing the former have an impact on the latter?
> ### Response to comment 6:
> We have examined the effect of using CPNG on spike frequency under different datasets, and the results are as follows (VGG16, ResNet19, VGG11). The use of CPNG may slightly increase the fire rate in the VGG structure, whereas in the ResNet structure, the fire rate will be slightly lower in the middle layers and slightly higher in the front and rear layers.
>
> Layers |  0 |  1 |  2 |  3 |  4 | 5 |  6 |  7 |  8 |  9 |  10 | 11 |  12 |  13 | 14
> |  ----  | ----  |  ----  | ----  |  ----  | ----  |  ----  | ----  |  ----  | ----  |  ----  | ----  |  ----  | ----  |  ----  | ----  |
> no CPNG  | 0.208 | 0.104 | 0.117 | 0.072 | 0.075 | 0.057 | 0.041 | 0.024 | 0.015 | 0.013 | 0.015 | 0.015 | 0.061 | 0.052 | 0.035
> CPNG      | 0.215 | 0.110 | 0.114 | 0.076 | 0.081 | 0.062 | 0.045 | 0.033 | 0.023 | 0.018 | 0.022 | 0.022 | 0.076 | 0.099 | 0.062
>
> Layers |  0 |  1 |  2 |  3 |  4 | 5 |  6 |  7 |  8 |  9 |  10 | 11 |  12 |  13 | 14 |15 | 16 | 17
> |  ----  | ----  |  ----  | ----  |  ----  | ----  |  ----  | ----  |  ----  | ----  |  ----  | ----  |  ----  | ----  |  ----  | ----  | ----  |  ----  | ----  |
> no CPNG  | .236 | .089 | .136 | .070 | .123 | .049 | .117 | .082 | .101 | .037 | .083 | .050 | .080 | .049 | .026 | .031 | .037 | .042
> CPNG       | .245 | .093 | .084 | .065 | .087 | .049 | .098 | .056 | .047 | .030 | .047 | .029 | .051 | .019 | .024 | .013 | .037 | .044
>
> Layers |  0 |  1 |  2 |  3 |  4 | 5 |  6 |  7
> |  ----  | ----  |  ----  | ----  |  ----  | ----  |  ----  | ----  | ----  |
> no CPNG  | 0.105 | 0.057 | 0.057 | 0.051 | 0.039 | 0.031 | 0.042 | 0.048
> CPNG     | 0.106 | 0.060 | 0.062 | 0.056 | 0.044 | 0.036 | 0.044 | 0.055
>
> ### Comment 7:
> It would be good either to increase the number of steps on the same tasks, or to directly apply it to tasks that involve longer sequences such as speech for instance.
> ### Response to comment 7:
> We did not consider this when completing this work because the previous work for deep SNN was focused on image classification. Here we give an example of voice recognition to demonstrate the effectiveness of our method. We extracted data with 100 time steps after processing the SHD dataset with a 10 ms interval. We trained it on a simple network (FC-700-->FC-256-->FC-20) with final results of 79.24\% (with CPNG) and 76.86\% (w/o CPNG).

---

> > ### Comment · Reviewer_z3Wm · 2022-11-18
> > **Response**
> >
> > Thank you for responding to the comments.  I think the response is appropriate.  However, I don't feel any need to alter my own ranking right now given that the other reviews err more negatively.

---

### Comment · Area_Chair_vLoN · 2022-11-18
**Please respond to author rebuttals**

Dear Reviewers,
The authors have submitted their rebuttals. Please have a look and respond to their efforts. This will be a respect to their hard work. Many thanks!

Area Chair

---

### Decision · Program_Chairs · 2023-01-20

**Decision:**

Reject

**Justification For Why Not Higher Score:**

It is surely not good enough for ICLR.

**Justification For Why Not Lower Score:**

N/A

**Metareview: Summary, Strengths And Weaknesses:**

The paper proposes a technique that can enhance the performance of the surrogate gradient (SG) method in training spiking neural networks. The proposed technique, called CPNG, adaptively adjusts the "effective domain" and shape of SGs based on the mean and variance statistics of the membrane potentials. The paper originally got one 6 (marginally above the threshold), one 5 (marginally below the threshold) and two 3s (reject). The major challenges include unconvincing or even unfair baseline comparison, unsatisfactory paper writing, incomplete literature review, limited novelty, etc.. Specifically, the Dspike method also dynamically changes the shape and high-value domain of the SGs, and shares similar performance with CPNG. The difference is that Dspike uses the finite difference to guild the shape-changing, and such a methodology is more reasonable and explainable. It is true that CPNG takes less overhead, but the overhead is not a very big issue, since both CPNG and Dspike cannot be implemented on neuromorphic hardwares, but only on GPUs. Furthermore, the finite difference used by Dspike can be efficiently approximated by sampling to achieve less overhead.

The authors responded actively to the review comments. Two reviewers raised his/her score to 6, but regretfuly reviewer eFt1 was still unsatisfied with the rebuttal (his/her comments were invisible to the authors). During the email discussion with the reviewers, no reviewer wanted to champion the paper and three reviewers agreed to reject the paper.

By the current status of the paper, the AC deemed that the paper does not match the standard of ICLR and hence recommended rejection.

**Summary Of Ac-Reviewer Meeting:**

Although the average score 5.25 is in the range of scores, 5.25-6.25, for discussion suggested by the program chairs, the score is only rank 6th among all my papers. I had brief email discussion with the reviewers. Two of them replied that they did not object that the paper be rejected and one was strongly against it. Thus it is sure that the paper be rejected.